# Dietary Animal to Plant Protein Ratio Is Associated with Risk Factors of Metabolic Syndrome in Participants of the AHS-2 Calibration Study

**DOI:** 10.3390/nu13124296

**Published:** 2021-11-28

**Authors:** Bahar Azemati, Sujatha Rajaram, Karen Jaceldo-Siegl, Ella H. Haddad, David Shavlik, Gary E. Fraser

**Affiliations:** School of Public Health, Loma Linda University, Loma Linda, CA 92354, USA; srajaram@llu.edu (S.R.); kjaceldo@llu.edu (K.J.-S.); ehaddad@llu.edu (E.H.H.); dshavlik@llu.edu (D.S.); gfraser@llu.edu (G.E.F.)

**Keywords:** dietary animal protein, metabolic syndrome, Adventist Health Study-2

## Abstract

Background: Few research studies have focused on the effects of dietary protein on metabolic syndrome and its components. Our objective was to determine the relationship between the type of dietary protein intake and animal to plant (AP) protein ratio with metabolic syndrome and its components. Methods: This population-based study had a cross sectional design and conducted on 518 participants of the Adventist Health Study 2 (AHS-2) Calibration Study. Two sets of three dietary 24-h recalls were obtained six months apart. Anthropometric measures and biochemical tests were performed in clinics. Regression calibration models were used to determine the association of type of dietary protein with metabolic syndrome and its components (raised triglyceride, raised blood pressure, reduced high-density lipoprotein cholesterol (HDL), raised fasting blood glucose and increased waist circumference). Results: The likelihood of metabolic syndrome was lower in those with higher total dietary protein and animal protein intake (*p* = 0.02).Total protein (β = 0.004, [95%CI: 0.002, 0.007]), animal protein intake (β = 0.004, [95%CI: 0.001, 0.007]) and AP protein intake ratio (β = 0.034, [95%CI: 0.021, 0.047]) were positively associated with waist circumference. Higher AP protein ratio was related to higher fasting blood glucose (β = 0.023, [95%CI: 0.005, 0.041]). Conclusion: Our study suggests that considering a significant amount of plant protein as a part of total dietary protein has beneficial effects on cardiometabolic risk factors.

## 1. Introduction

Metabolic syndrome has been recognized worldwide as a public health concern as it is associated with increased risk of cardiovascular disease (CVD), stroke, kidney diseases, and type 2 diabetes mellitus [1,2]. The syndrome involves a cluster of disorders identified as abdominal obesity, elevated triglycerides, low HDL, hypertension and increased fasting glucose [3]. Preventive strategies for metabolic syndrome include implementation of dietary and lifestyle changes for reducing fat mass, maintaining lean body mass, the management of hypertension, hyperglycemia and dyslipidemia. Dietary interventions have mainly focused on low glycemic load diets to favorably affect cardiometabolic features [4]. However, the types and amounts of fat and protein contents of these diets can also alter the outcomes [5].

Considering the effects of macronutrients on cardiometabolic health, dietary protein has received little attention compared with dietary fats and carbohydrates. Among various dietary patterns, adherence to vegetarian dietary patterns has been associated with a lower risk of developing metabolic syndrome abnormalities [6,7,8,9,10,11], diabetes [12,13,14] and CVD mortality [15]. However, the benefits of plant protein intake cannot be inferred from studies on vegetarians because other components of their diets such as other macronutrients, vitamins, minerals, phytochemicals, and fiber individually or in combination can influence cardiometabolic health outcomes [10].

The existing limited evidence on the various health effects of plant compared to animal proteins remains inconsistent. A large prospective cohort in the US demonstrated that higher plant protein intake was inversely associated with mortality from all CVD [15]. Dietary plant protein has been related to reduced CVD risk factors namely, lower blood pressure, improved lipid profiles and glucose control [16]. Epidemiological studies have focused on the unfavorable effects of protein rich animal-sources on the risk of elevated blood pressure and central obesity [17]. In a combined analysis of Nurses’ Health Study I & II and the Health Professionals Follow-up Study, the percentage of energy intake from plant protein was associated with a moderately decreased risk of type 2 diabetes [18]. It is worth noting that the results of observational dietary studies should be interpreted with caution since they may be confounded by other lifestyle habits and practices [19]. Despite these promising observations, recent reviews of interventional studies suggest favorable effects of soy protein with isoflavonoids, but not other plant proteins, on lipid profile [16]. Moreover, no differential effect on blood pressure, blood glucose or insulin was reported when dietary plant protein is directly compared with animal protein intake [16].

Although the benefits of plant based and vegetarian dietary patterns are documented [20,21,22], the overall benefits of plant protein on metabolic risk factors remains disputed. Our study was conducted to help fill this knowledge gap. Participants of the AHS-2 were quite a homogenous population in that they tend to be non-smokers, consume little alcohol, have lower body weights and generally have healthful diets [23]. With a large number of vegans and vegetarians in the cohort, plant protein shows a wide range of intake between 14–80 g/d. The animal to plant protein intake ratio has a range of 0 to 1.77 [24], whereas that of the US population is 1.8 to 2.2 [25]. Therefore, the purpose of this study was to examine the relationship between total protein and the type of dietary protein intake (animal and plant protein) with metabolic syndrome and its components in healthy individuals who consume a wide range of dietary protein and mostly a plant-based diet. We hypothesized that there would be an inverse association between plant protein intake and metabolic syndrome and or its components.

## 2. Materials and Methods

Study design and participants: This was a cross sectional study of the subsample of the AHS-2 participants from the calibration sub-study from whom detailed dietary, biochemical and physical activity data were obtained [26]. AHS-2 is a prospective population based study with 96,000 Seventh-day Adventists from the U.S. and Canada. The main goal of this study was to examine the association between lifestyle factors and some cancers. Details of the cohort and protocol of the study has been described elsewhere [26]. Information related to lifestyle and health, dietary patterns and physical activity were obtained through a comprehensive self-administered questionnaire. The calibration sub-study participants were randomly selected from the parent cohort by church and then subject-within church [27]. For this study, subjects who consumed alcohol, were current smokers, had an energy intake of less than 500 kcal or more than 4500 kcal and had incomplete data on dietary intake, demographic variables, components of metabolic syndrome, and physical activity were excluded from the analysis. Our final cohort consisted of 550 subjects.

Dietary assessment: Six 24-h dietary recalls were obtained from each subject using two sets of three (Saturday, Sunday, weekday) recalls which were taken six months apart. Nutrient analysis and data extraction were accomplished using the NDS-R 208 database. (version 4.06 or 5.0, Minneapolis, MN, USA). We estimated mean daily nutrients intake by using two sets of 24-h dietary recalls (Saturday intake + Sunday intake + 5 × weekday intake)/7. We used the regression residual method to adjust protein intake for total energy intake [28].

The Institutional Review Board of Loma Linda University approved the study protocol. At the time of enrollment, written consent was obtained from each participant.

Anthropometric and activity measures: A Tanita BF-350 (Tanita UK Ltd., Middlesex, UK) was used to measure participants’ weight. Weight was rounded up to the nearest 0.1 kg. Height was measured without shoes using a Seca 214 portable Height Rod (Seca corp, Hamburg, Germany). It was rounded up to the nearest quarter centimeter. Waist circumference was measured three times with an anthropometric tape 2.54 cm above the navel. The average of three values was used for the analyses. Physical activity (hour/day) was estimated via physical activity questionnaire [29]. Total physical activity was considered as the sum of mild, moderate, vigorous, and extremely vigorous activity. Subjects were also interviewed by phone and recalls for the past seven days were obtained to validate measures of physical activities and sedentary times.

Assessment of metabolic syndrome: Fasting blood glucose and cholesterol, low density lipoprotein cholestrol (LDL), HDL, and triglycerides concentrations were obtained using the Cholestech LDX System (Cholestech, Hayward, CA, USA) [30]. Blood pressure was measured three times using Omron Automatic Digital Blood Pressure Monitor HEM-747IC (Omron Healthcare, Inc., Vernon Hills, IL, USA) and the average of three values was used for analyses. The occurrence of metabolic syndrome was defined as the presence of three out of five components of metabolic syndrome, defined according to National Cholesterol Education Program Adult Treatment Panel III the 2009 Joint Scientific Statement. The five components are defined as follows: (1) Waist circumference >102 cm (in men) and >88 cm (in women); (2) Fasting plasma glucose >100 mg/dL or drug treatment for elevated glucose; (3) Blood pressure >130/85 mm Hg or antihypertensive drug treatment in a patient with a history of hypertension; (4) Triglycerides >150 mg/dL or drug treatment for elevated triglycerides; and (5) HDL <40 mg/dL (in men) and <50 mg/dL (in women) or drug treatment for reduced HDL [31].

### Statistical Analyses

Characteristics of participants according to metabolic syndrome were calculated by independent samples t-test and Chi-squared test. To correct for within subject variability and measurement error bias in the two sets of repeated 24-h dietary recalls, regression calibration was performed [32]. When it was appropriate, we used log-transformed (log 10) variables. In our multiple logistic regression models, we considered dietary protein (total, animal, plant, and animal to plant protein (AP ratio) as independent variables and metabolic syndrome as dependent variable. In these models we controlled for age, sex, ethnicity, physical activity, education, polyunsaturated fatty acids to saturated fatty acids Ratio (PS ratio), glycemic load, Body Mass Index (BMI), energy, and for the other types of protein that was not the independent variable. To assess the relationship between dietary protein and components of metabolic syndrome, we used dietary protein as independent variables and component of metabolic syndrome as dependent variable in our multiple linear regression models. For each component of metabolic syndrome, we adjusted for relevant confounders. Beta coefficients refer to degree of change in risk factors of metabolic syndrome per a gram per day increase in dietary protein intake. Statistical analysis was performed using Data Analysis and Statistical Software (STATA, version 12). *p* values less than 0.05 were considered as significant.

## 3. Results

A total of 518 subjects were selected from the participants of the calibration study. The mean age of participants was 60.2 ± 13.7 years. The results of the univariate analyses showed that those with metabolic syndrome had higher BMI (31.0 ± 6.5 vs. 27.0 ± 5.9, *p* < 0.0.001) and intake of trans fatty acids intake (*p* = 0.01) and lower intake of potassium (*p* = 0.01) (Table 1).

Based on regression analysis, we found that total dietary protein intake and animal protein intake (*p* = 0.02) were associated with 6% lower odds for metabolic syndrome Associations with plant protein did not reach significance. (Table 2).

Waist circumference was positively associated with total protein intake (β = 0.004, [95%CI: 0.002, 0.006]) animal protein (β = 0.00, [95%CI: 0.001, 0.007]) and AP ratio (β = 0.34, [95%CI: 0.021, 0.047]).

In other word, we observed 1.00 cm change in waist circumference per gram increase in total protein intake or animal protein consumption. In addition, 1.03 cm change in waist circumference per one unit increase in AP ratio was reported. HDL, triglycerides, systolic or diastolic blood pressure were not significantly related to dietary protein intake. However, fasting blood glucose was strongly associated with AP ratio (β = 0.023, [95%CI: 0.005, 0.041] or for each unit increase in AP ratio, fasting blood glucose was increased by 1.02 mg/dL. (Table 3).

## 4. Discussion

The aim of the present study was to compare the effects of total, plant and animal protein intake on metabolic syndrome and its components in a representative sample of adult and elderly participants of the AHS-2 Calibration study which includes many vegetarians and vegans. Our findings oppose our hypothesis; we found a lower likelihood of metabolic syndrome with a total dietary protein intake or animal protein consumption. Total protein, animal protein and AP protein ratio were associated with higher waist circumference. Relative amounts of animal to plant protein intake was also related to high fasting blood glucose.

In the present study total protein and animal protein intake were associated with lower likelihood of metabolic syndrome. The association between animal protein intake and metabolic syndrome is controversial. Our finding is in accordance with results of a longitudinal study which demonstrated that increment in the frequency of total and animal protein intake was associated with lower odds of metabolic syndrome [33]. However, it contrasts with results of a longitudinal study which showed an increased odds ratio for incident metabolic syndrome in those with the highest versus the lowest quartile of percent energy as total protein and animal protein, while the likelihood of metabolic syndrome was lower for dietary plant protein intake [17]. In another study, total dietary protein showed no association with incident metabolic syndrome in healthy adults [34]. The high proportion of vegetarians (53.2%) [35] in the AHS-2 sample with a habitual lower intake of total and animal-source protein may have confounded the effect of protein intake on metabolic syndrome [24]. The average total protein intake in AHS-2 participants is 67 g/d in men and 52 g/d in women, while that of US population is 90 g/d in men and 67 g/d in women, respectively [36]. The average percentages of protein intake derived from animal and plant protein sources of AHS-2 participants are ∼34% and 65%, respectively, whereas those of the US population are 62% and 30%, respectively [24]. It is worth noting that dietary protein from various food sources does not show similar relationships with metabolic syndrome. Chicken [17] and red meat intake have shown a positive association with incidence of metabolic syndrome [17,33,37], while fish [38], dairy [39] legumes and nuts [17] have revealed a negative association with metabolic syndrome. Although, dietary plant protein intake of AHS-2 is almost two times of that of the US population, the absolute amount of dietary plant protein was not associated with metabolic syndrome in this population which highlights the necessity of considering the effects of not just the quantity of dietary plant protein but also the different food sources and the total quantity of protein intake on metabolic syndrome or its components [16]. Our study demonstrates that the total dietary protein intake can lower the likelihood of metabolic syndrome when dietary plant protein is its major component.

As for the effects of protein intake on metabolic syndrome components, our data shows that although animal protein was not associated with metabolic syndrome, those with higher animal protein intake and those with a higher ratio of animal to plant protein had a higher waist circumference. A study of elderly Koreans with just one third of their total protein intake coming from animal sources, reported reduced waist circumference as daily intake of total protein, animal protein and plant protein increased. The Korean cohort was older and the average intake of total, animal and plant protein was lower than that of our study. This suggests that a certain required amount of protein is needed for maintaining normal body weight and waist circumference, hence up to this threshold only quantity of proteins if beneficial. However, higher intakes of proteins especially when a higher percent comes from animal source could be detrimental to body weight [40]. Generally longitudinal studies show increased body weight, abdominal fat and waist circumference with an increasing intake of dietary animal protein [41,42,43], with some studies reporting a decrease in those markers with intake of plant protein [44].

In the current study, neither total, animal or plant proteins were associated with the lipid components including HDL and triglycerides. The Framingham Heart Study Offspring cohort showed no association between dietary protein intake and lipid profile when protein intakes was considered as grams per day. However, when protein was expressed as gram per kilogram body weight per day, protein intake was associated with favorable annualized changes in HDL and triglycerides but those benefits in the lipid profile were confounded by changes in body weight and BMI status [44]. In line with the Framingham study, the Tehran Lipid and Glucose Study demonstrated that higher intake of dietary protein was associated with favorable effects on the lipid profile [43]. Studies reveal the importance of considering both the source and the type of the dietary protein. A recent systematic review comparing plant with animal protein intake in relation to cardiometabolic risk concluded that soy protein with isoflavones, but not soy protein alone or other plant proteins, led to greater decreases in total and LDL compared with protein from animal-sources [16]. A meta-analysis of RCT trials showed that substitution of plant protein for animal protein decreased LDL, non-HDL, and apolipoprotein B [45,46]. It is unclear why in our study dietary plant protein intake did not show an inverse association with the lipid biomarkers when much evidence exits on the beneficial effect of vegetarian diets on blood lipid [6,11,12]. A possible explanation could be related to the combined effect from several components found in vegetarian diets in addition to plant protein such as soluble fiber, saponins, steroids, polyphenols, and phytates which have lipid modifying properties. It has been shown that changes in HDL and triglycerides may be influenced more by other dietary factors or body weight and not directly associated to the type of dietary protein [47].

In the current study, the AP protein ratio was positively associated with fasting blood glucose. This is somewhat consistent with our previous findings in this population which revealed that total, animal and the animal to plant protein ratio were associated with insulin resistance [24]. A number of meta-analyses of prospective cohort studies showed that total and animal protein intake is associated with higher risk of type 2 diabetes, while plant protein intake decreased the risk [48,49,50,51]. In a meta-analysis of interventional studies substituting plant protein for animal protein improved glycemic control [52]. Although the literature is limited, it has been demonstrated that not all protein rich animal-source foods have the same impact on the risk of type 2 diabetes, with the intake of red and processed meats [53] and eggs [54] increasing the risk, while fish [55] and low-fat dairy products [51] reducing the risk. The complex influence of various protein rich animal foods on disease risk warrants further investigation.

The current evidence on the association of dietary protein quantity and quality with blood pressure is conflicting, with our own observation showing no association. A cross sectional study revealed an inverse association between the intake of plant protein and blood pressure, particularly in people with hypertension, but no association for total and animal protein intake [55]. In a cohort study, the intake of total, animal and plant protein were not associated with blood pressure, however, protein from grains showed an inverse association [56]. Results from the Framingham Offspring study demonstrated that both animal and plant protein intakes were associated with reduced risk of elevated blood pressure [57]. A meta-analysis of interventional studies reported an inverse relation between dietary protein intake and blood pressure and no differential effect on blood pressure when animal protein was directly compared to plant protein [58]. It is unclear why plant protein did not show an association to blood pressure in the current study. Vegetarian diets have typically been linked to reductions in blood pressure [59]. The underlying hypothesis being that plant protein, with a low lysine to arginine ratio, reduces blood pressure by stimulating nitric acid production [60]. However, altering the dietary lysine to arginine ratio has not been shown to have an effect on vascular reactivity and blood pressure [61]. Even though the amounts of arginine intake might be high in vegetarians due to a high intake of plant protein originated from soy and nuts; these food items in addition to non-soy legumes are also rich in lysine, hence this issue needs to be explored. Moreover, plant protein is consumed along with other nutrients, so there is a possibility that the observed effects in various studies come from the cluster of nutrients and not just plant protein. Furthermore, in most observational studies adjustments were made for nutrients that are indicators of a healthy lifestyle (for example dietary fiber), incomplete adjustment for lifestyle or dietary factors may have resulted in residual confounding.

The biological mechanisms that might explain the adverse association between total and animal protein intake and an increased waist circumference and fasting blood sugar are not completely understood. Animal source high protein foods are energy-dense and often high saturated-fat foods that are associated with increased body weight and abdominal fat. A possible explanation for the differing risk associated with protein could be related to the different amino acid profiles of animal compared to plant proteins [62]. It has been well documented that branched chain amino acids (BCAA) are more abundant in animal protein and elevated BCAA have been reported in obese humans and animal models of obesity [63,64]. The dietary ingestion of BCAA and sulfur containing amino acids (SCAA), abundant in animal protein foods are also associated with fat mass and central obesity [65]. Differences in amino acid profiles have also been linked to glycemic control since BCAA and possibly glutamic acid [66] are known to activate mammalian target of rapamycin complex 1 (mTORC1) which in turn leads to insulin resistance [65]. However, it is not clear whether the high blood levels of BCAAs are a cause or an effect of increased weight and abdominal fat [65,66,67,68,69].

Inconsistency observed between epidemiological and interventional studies could be due to differences in study design, food consumption assessment methods and food group classifications used in the various studies, the duration of studies, health status of participants as well as adjustment for various confounding factors.

One of the strengths of our study was using multiple 24-h dietary recalls to assess dietary intake of subjects, which allowed for within-subject variability adjustment using regression calibration [70]. In addition, there was a wide range of animal and plant protein intake in this population; therefore, we were able to assess relationship between dietary protein intake from different sources and risk factors of metabolic syndrome. An additional strength of our study was assessing the aforementioned relationship without the effects of confounders such as smoking and alcohol use which enhances our internal validity; however, excluding these individuals from our sample limited external validity. It is worth mentioning that adjusting for covariates such as smoking, and alcohol consumption in the statistical models might still affect the association of dietary protein intake and cardiometabolic outcomes due to other factors relevant to these confounders. In the majority of the observational studies when one type of protein was the main predictor of the outcome, another type of protein was not adjusted for, but in the current study we considered this issue. Doing so allowed us to test the relationship of one type of protein unconfounded by another type of protein.

Our study also had several limitations. Because of the cross-sectional design, reverse causality may have occurred which can bias the results toward the null, and we cannot establish temporality of the associations. Since dietary and lifestyle practices of the AHS-2 participants may not be similar to those of the population [26], generalizing our results to general population should be made with caution. We believe that more research is needed to investigate the effects of animal protein rich food groups and different dimensions of nutritional profiles including polyphenols and amino acids compositions from diverse protein source on metabolic syndrome or its components in populations with different sources and levels of protein intake.

## 5. Conclusions

In conclusion, our study provides some evidence that the intake of higher animal protein and the relative amount of animal to plant protein is associated with cardiometabolic risk factors namely higher waist circumference and fasting blood glucose. However, plant protein intake had no effect on glucose hemostasis or body composition. Our study suggests that considering a significant amount of plant protein as a part of total dietary protein has beneficial effects on cardiometabolic risk factors. In addition, focusing on quantity as well as quality of the protein in order to reduce the cardiometabolic risk factors is indispensable. The main focus should be on meeting the adequate amount of protein, however, to ensure greater benefits and avoid potential detrimental effects, limiting animal protein sources while increasing a wide variety of plant-based proteins may be helpful.

## Figures and Tables

**Table 1 nutrients-13-04296-t001:** Subject characteristics according to metabolic syndrome status.

		Metabolic Syndrome		*p* Value
		No	Yes	
N		402	116	
Age		60.4 ± 14.0	57.6 ± 14.2	0.06
BMI		26.4 ± 5.4	32.5 ± 6.6	<0.001
Sex	Female	258 (64.1)	75 (64.6)	0.9
	Male	144 (35.8)	41 (35.3)	
Race	White	242 (60.2)	73 (62.9)	0.5
	Non-White	160 (39.8)	43 (37.0)	
Education	High school & lower	89 (22.3)	37 (31.9)	0.08
	Some college	139 (34.8)	32 (27.5)	
	Bsc & higher	171 (42.8)	47 (40.5)	
Physical activity (hrs/d)		36.5 ± 10.2	34.9 ± 10.0	0.1
Animal protein (g)		25.1 ± 19.4	28.9 ± 20.7	0.06
Plant protein (g)		34.6 ± 11.1	32.9 ± 11.0	0.1
AP ^1^ protein ratio		0.7 ± 0.8	0.9 ± 0.9	0.06
Total protein * (g)		59.7 ± 14.5	61.9 ± 5.1	0.1
GL ^2^		123.4 ± 42.7	115.1 ± 39.7	0.06
Unsaturated fatty acids * (g)		51.3 ± 1.8	52.1 ± 2.0	0.5
PS ratio ^3^		1.1 ± 0.4	1.0 ± 0.5	0.1
Trans fatty acids (g)		1.7 ± 0.7	1.9 ± 0.7	0.01
Calcium (mg)		990.0 ± 471.7	931.6 ± 479.6	0.2
Sodium (mg)		2472.2 ± 849.2	2455.0 ± 836.5	0.8
Potassium (mg)		2533.5 ± 829.8	2310.7 ± 29.4	0.01
Magnesium (mg)		368.6 ± 156.6	351.5 ± 179.1	0.3

Continuous variables are presented as Mean ± SD and categorical variables are shown as Count (%); ^1^ AP protein ratio; Animal to plant protein ratio; ^2^ GL; Glycemic load; ^3^ PS ratio; Polyunsaturated fatty acids to Saturated fatty acids ratio. * Dietary protein and fatty acids are energy adjusted.

**Table 2 nutrients-13-04296-t002:** Association between dietary protein intake and metabolic syndrome in AHS-2 calibration study (*n* = 518).

Dietary Protein Intake	Odds Ratio (95% CI) Crude	*p* Value	Odds Ratio (95% CI)	*p* Value
Total protein	0.99 (0.95, 1.02)	0.6	0.94 (0.89, 0.99)	0.02
Animal protein	0.98 (0.94, 1.02)	0.5	0.94 (0.89, 0.99)	0.02
Plant protein	0.96 (0.90, 1.04)	0.4	0.94 (0.85, 1.04)	0.2
AP protein	1.06 (0.84, 1.33)	0.5	0.94 (0.85, 1.04)	0.2

Adjusted for age, ethnicity, sex, physical activity, PS ratio, glycemic load, BMI, energy, type of dietary protein from other sources than the one under study, if applicable, total protein and AP protein ratio.

**Table 3 nutrients-13-04296-t003:** Association between dietary protein intake and components of metabolic syndrome in AHS-2 calibration study. (*n* = 518).

	Dietary Protein Intake							
	Total Protein		Animal Protein		Plant Protein		AP Protein Ratio	
Components of Metabolic Syndrome	β Coefficient (95% CI)	*p* Value	β Coefficient (95% CI)	*p* Value	β Coefficient (95% CI)	*p* Value	β Coefficient (95% CI)	*p* Value
Waist circumference (crude) *	0.005 (0.002, 0.007)	<0.001	0.004 (0.002, 0.007)	0.001	0.003 (−0.0008, 0.008)	0.1	0.035 (0.021, 0.049)	<0.001
Waist circumference ^1,^**	0.004 (0.002, 0.007)	<0.001	0.004 (0.001, 0.007)	0.006	0.002 (−0.002, 0.007)	0.3	0.034 (0.021, 0.047)	<0.001
High density lipoprotein (crude) *	−0.001 (−0.005, 0.002)	0.3	−0.001 (−0.006, 0.003)	0.4	−0.002 (-.010, 0.006)	0.6	−0.004 (−0.031, 0.023)	0.7
High density lipoprotein ^2,^**	0.001 (−0.002, 0.006)	0.4	0.002 (−0.002, 0.008)	0.3	0.006 (−0.005, 0.0.018)	0.2	0.012 (−0.037, 0.062)	0.6
Triglycerides (crude) *	0.001 (−0.005, 0.007)	0.7	−0.001 (−0.009, 0.006)	0.6	−0.009 (−0.022, 0.004)	0.1	0.038 (−0.005, 0.082)	0.08
Triglycerides ^3,^**	−0.001 (−0.007, 0.004)	0.6	−0.002 (−0.010, 0.004)	0.4	−0.007 (−0.020, 0.004)	0.2	0.009 (−0.033, 0.052)	0.6
Fasting blood glucose (crude) *	0.004 (0.002, 0.007)	<0.001	0.004 (0.001, 0.007)	0.006	0.003 (−0.001, 0.009)	0.1	0.038 (0.022, 0.055)	<0.001
Fasting blood glucose ^4,^**	0.001 (−0.0009, 0.004)	0.1	0.001 (−0.001, 0.004)	0.3	0.0006 (−0.004, 0.005)	0.8	0.023 (0.005, 0.041)	0.009
Systolic blood pressure (crude) *	−0.06 (−0.295, 0.194)	0.6	−0.025(−0.329, 0.279)	0.8	0.036 (−0.486, 0.559)	0.8	−0.878 (−2.54, 0.783)	0.2
Systolic blood pressure ^5^	−0.570 (−1.445, 303)	0.2	−0.752 (−2.850, 1.344)	0.4	−0.984 (−4.655, 2.685)	0.5	−2.85 (−6.325, 0.615)	0.1
Diastolic blood pressure (crude) *	0.185 (0.044, 0.327)	0.01	0.137 (−0.028, 0.302)	0.1	0.018 (−0.262, 0.299)	0.8	1.72(0.857, 2.58)	<0.001
Diastolic blood pressure ^5^	0.106 (−0.299, 0.512)	0.6	−0.083 (−0.980, 0.812)	0.8	−0.325 (−1.913, 1.262)	0.6	1.47 (−0.540, 3.49)	0.1

^1^ Adjusted forage, ethnicity, sex, physical activity, PS ratio, glycemic load, energy, total protein and animal to plant (AP) protein ratio, and in addition, animal and plant protein intake was adjusted for intake of the other protein source.; ^2^ Adjusted for; age, ethnicity, sex, physical activity, waist circumference, PS ratio, trans fatty acids, energy and type of dietary protein from other sources than the one under study, if applicable, total protein and AP protein ratio.; ^3^ Adjusted for; age, ethnicity, sex, physical activity, waist circumference, glycemic load, unsaturated fatty acids and type of dietary protein from other sources than the one under study, if applicable, total protein and AP protein ratio.; ^4^ Adjusted for; age, ethnicity, sex, physical activity, glycemic load, waist circumference, energy and type of dietary protein from other sources than the one under study, if applicable, total protein and AP protein ratio.; ^5^ Adjusted for; age, ethnicity, sex, physical activity, glycemic load, waist circumference, PS ratio, calcium, magnesium, potassium, sodium, energy and type of dietary protein from other sources than the one under study, if applicable, total protein and AP protein ratio. * Adjusted for type of protein and energy. ** Variables are log transformed.

## Data Availability

The data presented in this study are available on request from the corresponding author. The data are not publicly available due to privacy of research participants.

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
