# Peer review of "Dietary Animal to Plant Protein Ratio Is Associated with Risk Factors of Metabolic Syndrome in Participants of the AHS-2 Calibration Study"

_nutrients, 2021, doi:10.3390/nu13124296_

Round 1

Reviewer 1 Report

Comments on the manuscript entitled Dietary animal protein intake is associated with risk 2 factors of metabolic syndrome in participants of the 3 AHS-2 Calibration Study
Increasing the proportion of plant-based protein sources in human diet is important both from health and environmental reasons. However, evidence-based knowledge of health effects of more plant-based diets and the mechanisms mediating their effects on human health is still limited and further research is needed. This manuscript is well written and provides new evidence of the effects of more plant-based diets on human health at population level.
My main concern considering the study is the method used to measure and calculate the nutrient intakes. Estimating mean daily nutrient intake by using two sets of 24-hour dietary recalls and calculating [(Saturday intake + Sunday intake + 5 × weekday intake) / 7] inevitably attaches more weight on the food consumption during the weekend, which is known to be different and often unhealthier than food consumption on weekdays. Furthermore, food intake on Monday and Friday are also known to be different than food intake on other weekdays. Food intake of one weekday cannot be used to represent the intake of other weekdays and multiplying it by five in the equation (as done in the current study) distorts the result even more. Using 4-day food records or recalls including three weekdays and one weekend day is a widely accepted method for collecting data on food intake. For resource reasons, 3-day records/recalls can also be used, but again including two weekdays and one weekend day, and calculating the daily mean based on those days.
Minor comments on the manuscript:
Rows 38-39: “Dietary interventions have mainly focused on low glycemic load diets to favorably affect cardiometabolic features.” Provide some references to support this claim.
Rows 64-66: “Participants of the Adventist Health Study-2 (AHS-2) are unique in that they tend to be non-smokers, consume little alcohol, have lower body weights and generally have healthful diets thus lessening confounding from lifestyle factors.” These all are lifestyle factors – as the authors also mention in the discussion in relation to the protein intake (discussion, rows 14-16). However, I would rather say here that the participants were quite a homogenous population, but I suggest removing the mention “lessening confounding from lifestyle factors”.
Row 73: A full stop is lacking.
Row 77: A writing error: “based” instead of “base”
Row 133-134: “Those with metabolic syndrome were mostly Non-Whites, males with high school or lower education.” The data and analyses to support this statement have not been presented – where does this result come from? As far as I can judge, this result cannot be seen from the tables. According to the table 1, there are no differences between the subjects with and without the metabolic syndrome in sex, race, or education. Please show the analyses supporting the statement or revise the text.
Discussion
Row 8: A full stop is lacking.
Rows 14-16: “The high proportion of vegetarians in the AHS-2 sample 14 with a habitual lower intake of total and animal-source protein may have confounded the effect of 15 protein intake on metabolic syndrome.” How high was the proportion of vegetarians in the study population? This is
an interesting thing and should be mentioned here, even though the result would have been published earlier. And unlike you mention in the introduction, vegetarian diet does not reduce confounding from lifestyle factors (see also my previous comment above).
Rows 57-59: “It is unclear why in our study dietary plant protein intake did not show an inverse association with the lipid biomarkers when much evidence exits on the beneficial effect of vegetarian diets on blood lipid.” One of the reasons for this may be the uncertainty in measuring the food intake; the result might have been different if the food intake on weekdays would have been measured more reliably.
Rows 62-64: “Unlike other lipids, changes in HDL and triglycerides may be influenced more by other dietary factors or body weight and not directly associated to the type of dietary protein.” This is true, but in this context the sentence is a bit confusing; it can be interpreted so that you found some changes in lipids other than HDL and triglycerides in this study, even though you did not. Overall, it is not surprising, since it is very difficult to change HDL and triglycerides by dietary means; alcohol consumption may increase them both, but otherwise they are other lifestyle factors that affect HDL and triglycerides.
Row 71-72: “…not all protein rich animal-source foods have the same impact on the risk of type 2 diabetes”. The same should also be discussed in relation to plant protein sources (soy, legumes other than soy, nuts and seeds, cereals) – they may also have different effects on T2D.
Rows 77-79: “A cross sectional study revealed an inverse association between the intake of plant protein, particularly in people with hypertension, and blood pressure but no association for total and animal protein intake.” Language revision needed: ... between the intake of plant protein and blood pressure, particularly in people with hypertension, but...
Rows 85-87: “It is unclear why plant protein did not show an association to blood pressure in the current study. Vegetarian diets have typically been linked to reductions in blood pressure [57].” Here again the reason for this may be the unreliability of measuring the food intake.
Row 101: “…might have a negative effect on body weight and abdominal fat.” It would be better to say that they often are associated with increased body weight and abdominal fat. The sentence in its current form may be misleading.
Row 115-116: “One of the strengths of our study was using multiple 24-h dietary recalls to assess dietary intake of subjects…” There is nothing wrong in the 24-h dietary recall as such. However, you should discuss measuring the nutrient intake not as a strength but a weakness of the study, due to the strong emphasizing on the intake during weekend days.
Rows 117-119: “In addition, there was a wide range of animal and plant protein intake in this population; therefore, we were able to assess relationship between dietary protein intake from different sources and risk factors of metabolic syndrome.” The results on animal and plant protein intakes should be shown at least in the supplementary material or even in the article, if they have not been published before. Furthermore, did you analyse the consumption of different sources of dietary protein on food group level (meat and meat products, dairy products, bread and cereals, legumes, etc.)? It would also be a result worth presenting!

Author Response

Comments on the manuscript entitled Dietary animal protein intake is associated with risk 2 factors of metabolic syndrome in participants of the 3 AHS-2 Calibration Study

Increasing the proportion of plant-based protein sources in human diet is important both from health and environmental reasons. However, evidence-based knowledge of health effects of more plant-based diets and the mechanisms mediating their effects on human health is still limited and further research is needed. This manuscript is well written and provides new evidence of the effects of more plant-based diets on human health at population level.

My main concern considering the study is the method used to measure and calculate the nutrient intakes. Estimating mean daily nutrient intake by using two sets of 24-hour dietary recalls and calculating [(Saturday intake + Sunday intake + 5 × weekday intake) / 7] inevitably attaches more weight on the food consumption during the weekend, which is known to be different and often unhealthier than food consumption on weekdays. Furthermore, food intake on Monday and Friday are also known to be different than food intake on other weekdays. Food intake of one weekday cannot be used to represent the intake of other weekdays and multiplying it by five in the equation (as done in the current study) distorts the result even more. Using 4-day food records or recalls including three weekdays and one weekend day is a widely accepted method for collecting data on food intake. For resource reasons, 3-day records/recalls can also be used, but again including two weekdays and one weekend day, and calculating the daily mean based on those days.

We respectfully disagree with the statement that our weighting scheme gives more weight to the weekends.  It is in fact the correct way to provide an unbiased estimate of the week’s intake—admittedly under the hypothesis that the weekdays are all, on average, the same as each other.  We did treat Saturday and Sunday separately and obtained two estimates of each, widely separated in time. As an example, say for some particular food that the intakes were (in some units) Saturday 20; Sunday 10, random weekday 30.  Let the actual weekdays (mostly unmeasured) be 20, 40, 50, 10, 30. Assume that the randomly chosen weekday was on average representative of the 5 weekdays, hence would on average take a value of 30. Then our synthetic week total is 5x30 + 20+10=180, which of course is exactly the total of the actual underlying week.  We are not aware of information that Monday and Friday are systematically different, certainly as would pertain to this population. Even if there are minor differences, this is surely much improved over a common previous practice of assuming that 1-2 random days are unbiased estimates of a whole diet.

We agree that the weekdays will contribute more to the variance of the weekly estimate than the weekends, but that should not affect the validity of our regression results”.

Minor comments on the manuscript:

Rows 38-39: “Dietary interventions have mainly focused on low glycemic load diets to favorably affect cardiometabolic features.” Provide some references to support this claim.

Revised.
Rows 64-66: “Participants of the Adventist Health Study-2 (AHS-2) are unique in that they tend to be non-smokers, consume little alcohol, have lower body weights and generally have healthful diets thus lessening confounding from lifestyle factors.” These all are lifestyle factors – as the authors also mention in the discussion in relation to the protein intake (discussion, rows 14-16). However, I would rather say here that the participants were quite a homogenous population, but I suggest removing the mention “lessening confounding from lifestyle factors”.

Revised

Row 73: A full stop is lacking.

Revised

Row 77: A writing error: “based” instead of “base”

Revised

Row 133-134: “Those with metabolic syndrome were mostly Non-Whites, males with high school or lower education.” The data and analyses to support this statement have not been presented – where does this result come from? As far as I can judge, this result cannot be seen from the tables. According to the table 1, there are no differences between the subjects with and without the metabolic syndrome in sex, race, or education. Please show the analyses supporting the statement or revise the text.

Revised

Discussion

Row 8: A full stop is lacking.

Revised

Rows 14-16: “The high proportion of vegetarians in the AHS-2 sample 14 with a habitual lower intake of total and animal-source protein may have confounded the effect of 15 protein intake on metabolic syndrome.” How high was the proportion of vegetarians in the study population? This is an interesting thing and should be mentioned here, even though the result would have been published earlier.

Revised

And unlike you mention in the introduction, vegetarian diet does not reduce confounding from lifestyle factors (see also my previous comment above).

My assumption is that those who are following vegetarian diets are more likely to be health conscious and comply with healthier lifestyle.

Rows 57-59: “It is unclear why in our study dietary plant protein intake did not show an inverse association with the lipid biomarkers when much evidence exits on the beneficial effect of vegetarian diets on blood lipid.”

One of the reasons for this may be the uncertainty in measuring the food intake; the result might have been different if the food intake on weekdays would have been measured more reliably.

Addressed in response to the main concern.

Rows 62-64: “Unlike other lipids, changes in HDL and triglycerides may be influenced more by other dietary factors or body weight and not directly associated to the type of dietary protein.” This is true, but in this context the sentence is a bit confusing; it can be interpreted so that you found some changes in lipids other than HDL and triglycerides in this study, even though you did not. Overall, it is not surprising, since it is very difficult to change HDL and triglycerides by dietary means; alcohol consumption may increase them both, but otherwise they are other lifestyle factors that affect HDL and triglycerides.

Revised

Row 71-72: “…not all protein rich animal-source foods have the same impact on the risk of type 2 diabetes”. The same should also be discussed in relation to plant protein sources (soy, legumes other than soy, nuts and seeds, cereals) – they may also have different effects on T2D.

I do agree with you. However, it has been shown that overall vegetable protein intake has protective effect on T2D.

Rows 77-79: “A cross sectional study revealed an inverse association between the intake of plant protein, particularly in people with hypertension, and blood pressure but no association for total and animal protein intake.” Language revision needed: ... between the intake of plant protein and blood pressure, particularly in people with hypertension, but...

Revised

Rows 85-87: “It is unclear why plant protein did not show an association to blood pressure in the current study. Vegetarian diets have typically been linked to reductions in blood pressure [57].” Here again the reason for this may be the unreliability of measuring the food intake.

Addressed in response to the main concern.

Row 101: “…might have a negative effect on body weight and abdominal fat.” It would be better to say that they often are associated with increased body weight and abdominal fat. The sentence in its current form may be misleading.

Revised

Row 115-116: “One of the strengths of our study was using multiple 24-h dietary recalls to assess dietary intake of subjects…” There is nothing wrong in the 24-h dietary recall as such. However, you should discuss measuring the nutrient intake not as a strength but a weakness of the study, due to the strong emphasizing on the intake during weekend days.

Addressed in response to the main concern.

Rows 117-119: “In addition, there was a wide range of animal and plant protein intake in this population; therefore, we were able to assess relationship between dietary protein intake from different sources and risk factors of metabolic syndrome.” The results on animal and plant protein intakes should be shown at least in the supplementary material or even in the article, if they have not been published before.

It has been shown in Table 3.

Furthermore, did you analyse the consumption of different sources of dietary protein on food group level (meat and meat products, dairy products, bread and cereals, legumes, etc.)? It would also be a result worth presenting!

No. We did not analyze the consumption of dietary protein on food group level.

Reviewer 2 Report

The Authors in the manuscript have found that dietary protein from plant source has a beneficial effects on metabolic syndrome. But the details mechanism behind this was not clearly explained in the manuscript.

Author Response

The Authors in the manuscript have found that dietary protein from plant source has a beneficial effect on metabolic syndrome. But the details mechanism behind this was not clearly explained in the manuscript.

Please note that epidemiological and interventional studies attempted to evaluate the benefits of two types of proteins on CVD, but a strict separate impact is difficult to prove because of the natural mix of dietary proteins people usually consume in addition to the effect of other non-protein compounds on CVD risk factors.

We did not state that "dietary protein from plant source has a beneficial effect on metabolic syndrome". Based on our findings the intake of higher animal protein and the relative amount of animal to plant protein is associated with cardiometabolic risk factors namely higher waist circumference and fasting blood glucose and the possible mechanisms were discussed.

We suggested that " a significant amount of plant protein as a part of total dietary protein has beneficial effects on cardiometabolic risk factors". The possible mechanism of these effects could be due to the amino acid composition of plant protein e.g the ability of arginine to decrease insulin resistance, to decrease glycation end products formation, to increase NO production, and decrease angiotensin II levels and oxidative stress.

Reviewer 3 Report

Dear authors thank you for your huge work. I found some weaknesses in your work:

  • conclusion are not original. The results about lower likelihood and total protein intake could be widely discussed. You could do some hypothesis about your findings
  • have you got data about other participant comorbidities?
  • I think you article could be more fascinating using some figures or graphics
  • some concept are repetitive. You could shorten your paper and make it more fluid.

Best regards

Author Response

Dear authors thank you for your huge work. I found some weaknesses in your work:

  • conclusion are not original. The results about lower likelihood and total protein intake could be widely discussed. You could do some hypothesis about your findings.

In our study, a high proportion of total protein intake came from plant protein sources also the participants were low meat eaters (addressed in discussion section; row 14-20).

The possible mechanism of the favorable effects of plant proteins on metabolic syndrome could be due to the amino acid composition of plant protein, for example the ability of arginine to decrease insulin resistance, to decrease glycation end products formation, to increase NO production, and decrease angiotensin II levels and oxidative stress.

  • have you got data about other participant comorbidities?

The subjects in the AHS2 parent and its calibration study were healthy individuals living in the community with low comorbidities.

  • I think you article could be more fascinating using some figures or graphics

Our preference is to use tables to communicate the results.

  • some concept are repetitive. You could shorten your paper and make it more fluid.

It is not specified which concepts are repetitive to enable us to revise them. However, we have deleted some parts of the discussion (e.g Row 29-30).